# Prevalence and Characteristics of *Streptococcus agalactiae* from Freshwater Fish and Pork in Hong Kong Wet Markets

**DOI:** 10.3390/antibiotics11030397

**Published:** 2022-03-16

**Authors:** Dulmini Nanayakkara Sapugahawatte, Carmen Li, Priyanga Dharmaratne, Chendi Zhu, Yun Kit Yeoh, Jun Yang, Norman Wai Sing Lo, Kam Tak Wong, Margaret Ip

**Affiliations:** Department of Microbiology, Faculty of Medicine, Prince of Wales Hospital, The Chinese University of Hong Kong, Sha Tin, Hong Kong SAR, China; dulmini87@hotmail.com (D.N.S.); 2carmen.li@cuhk.edu.hk (C.L.); priyanga@cuhk.edu.hk (P.D.); samzhu@cuhk.edu.hk (C.Z.); yeohyunkit@cuhk.edu.hk (Y.K.Y.); junyang@link.cuhk.edu.hk (J.Y.); normanlo@cuhk.edu.hk (N.W.S.L.); kamtakwong@cuhk.edu.hk (K.T.W.)

**Keywords:** freshwater fish, pig, *Streptococcus agalactiae*, Group B *Streptococcus*, aquaculture, One Health, antimicrobial resistance, WGS, multidrug resistance

## Abstract

We report the antimicrobial resistance of 191 fish and 61 pork Group B *Streptococcus* (GBS) procured from Hong Kong wet markets. Two-hundred-and-fifty-two GBS strains were isolated from 992 freshwater fish and 361 pig offal during 2016–2019. The strains were isolated from homogenised samples and plated on selective media, followed by identification through MALDI-TOF-MS. Molecular characterisation, an antibiotic susceptibility test, and biofilm formation were performed on the strains. The isolation rates of the fish GBS and pig GBS were 19.3% (191 strains from 992 freshwater fish) and 16.9% (61 strains from 361 pig organs), respectively. The fish GBS was predominantly serotype Ia, ST7, while pig GBS was serotype III, ST651 (45 strains). An antibiotic susceptibility test revealed that the fish GBS were mostly antibiotic-sensitive, while the pig GBS were multidrug-resistant. A biofilm formation experiment showed that over 71% of fish GBS and all pig GBS had moderate biofilm formation ability. In general, the prevalence rate of GBS in animals and the multidrug resistance phenotype presented in the strains raise concerns about its zoonotic potential and effects on public health.

## 1. Introduction

*Streptococcus agalactiae*, also known as Group B *Streptococcus* (GBS), is a gram-positive bacterium that belongs to Lancefield group B. The organism was first described in 1887 but was only reported as a human pathogen in 1935 [1]. Although it is a part of the normal microbiota in the human gastrointestinal tract and the lower genital tracts of women, GBS is also known as a causative organism of life-threatening septicemia in humans, especially in newborns and pregnant women, as well as of invasive infections in non-pregnant adults [2]. It has a broad host range, from fishes, reptiles, and amphibians to mammals, such as cows, in which it causes bovine mastitis [3].

Pathogenic GBS has hindered global aquaculture activity due to its association with instances of severe die-off in farmed fish [4]. In recent years, streptococcosis has been reported in food animal species, including Asian bighead carp (*Hypophthalmichthys nobilis*), snakehead fish (*Channa* spp.), and tilapia (*Oreochromis* sp.) [5], as well as cows [6]. Clinical episodes of bacteraemia and meningitis have been reported in Singapore in association with raw fish consumption, specifically serotype III, sequence type (ST) 283 [7,8]. GBS disease is currently endemic to several tilapia-producing countries in Southeast Asia, including Malaysia, Indonesia, Vietnam, and Thailand, which negatively affects the development of this business sector [8].

Asia has dominated the global aquaculture production of farmed aquatic animals for the last two decades. China has been the world’s biggest producer since 2018, claiming 57.93% (47.6 million tonnes) of the world’s production. In addition, India, Indonesia, Vietnam, Bangladesh, Egypt, Norway, and Chile have contributed to regional or world production to varying degrees in recent years [9]. China has also provided over 54 million metric tonnes of pork worldwide, supporting 50% of the global demand [10]. The increased production of and demand for freshwater fish and pork may potentiate the risk of the emergence of zoonotic bacterial infections, such as GBS ST283 [8]. If neglected, with its broad host range and potential to cause host mortality, zoonotic GBS can jeopardize public health, as well as that of the animal farming industry.

In Hong Kong, fresh food products are often purchased from traditional wet markets. Currently, 97 public wet markets are functioning across Hong Kong’s 18 districts that are managed by the Food and Environmental Hygiene Department of Hong Kong SAR [11]. The wet markets in Hong Kong also sell fresh meat, including poultry, beef, and pork, which is supplied from three licensed slaughterhouses for retail. Animal parts of cows and pigs, such as offal, heads, tails, and chicken feet, are also available at these markets due to their use in local cuisine [12]. However, only specific markets sell live poultry, and interventions are in place to minimize zoonotic influenza transmission [13]. In addition, live seafood can be kept in “aquariums” before being sold.

As East Asian countries are significant producers of freshwater fish, and the consumption of raw fish is common in the region, we hypothesize that there is the potential for zoonosis to be acquired from fish, as in the Singapore outbreak [8]. Furthermore, given the complex food habits among the Hong Kong population in addition to its consumption of freshwater fish, there might be some other food sources, such as pork, that can also be contaminated with GBS and passed onto humans through meat processing chains. Therefore, our present study sought to isolate and characterize GBS carriage from freshwater fishes and pig organs procured from wet markets across Hong Kong from mid-2016 to 2019 by phenotypic and molecular methods.

## 2. Results

### 2.1. Prevalence of GBS in Food Animals and Its Climatic Association

In total, 252 GBS strains were isolated from 992 freshwater fish and 361 pig organs purchased in wet markets across all 18 districts of Hong Kong. The isolation rates in fish and pig offal were 19.3% (191 strains) and 16.8% (61 strains), respectively. The isolation rate was highest in tilapia (34.1%), followed by black carp (22.3%). Pig GBS (pGBS) was most prevalent in tongues (24.8%) and less observed in the small (3.2%) and large intestines (13%) (Table 1). To assess the factors contributing to the animal GBS isolation rate, we examined possible climatic associations in Hong Kong during our study period. We reviewed Hong Kong’s average seasonal temperature, humidity, and rainfall records during our collection period. The isolation of fish GBS (fGBS) peaked during the summer months (June–August), when the mean temperature, relative humidity, and rainfall were >28 °C, >81% and >400 mm (Figure 1). No such phenomenon was observed in pigs.

### 2.2. Antibiotic Susceptibility and Biofilm Formation in Fish and Pig GBS

Antibiotic susceptibility testing of 191 fGBS and 61 pGBS isolates showed distinct MIC results. A small cohort of fGBS was resistant to tetracyclines (*n* = 12, 4.7%), with 3.1% and 1.5% of strains resistant to erythromycin and clindamycin, respectively. One strain was resistant to penicillin and ciprofloxacin and reported elsewhere [14]. Multidrug resistance (MDR) was observed in four fish strains (2.1%) (Appendix A). By contrast, over 90% of pGBS were resistant to tetracyclines, 88.5% were resistant to erythromycin, and all but one GBS strain was resistant to clindamycin (Table 2). Multidrug resistance was observed in 56 out of 61 pig strains (91.8%) (Appendix A).

The biofilm formation test showed that 78.5% (198/252) of GBS strains were biofilm formers. Over 71% (137/191) of fGBS and 98.4% (60/61) of pGBS were moderate biofilm formers, while one pGBS isolate was a strong biofilm former (Appendix A). None of the fGBS strains were strong biofilm formers. However, 28% (54/191) of fGBS were weak or non-biofilm-forming (Appendix A). There was no significant difference in the biofilm formation.

### 2.3. Molecular Characteristics of Capsular Serotypes among Freshwater Fish and Pigs

Multiplex PCR revealed serotype Ia was dominant in fGBS (96.3%, 184/191) with a minority of serotypes III-NT (*n* = 1), V (*n* = 2) and NT (*n* = 4) (Table 3). One-hundred-and-eighty-seven representative strains encompassing all four serotypes (including serotype NT) underwent WGS. Sequence type (ST) 7 was prevalent in our fGBS (*n* = 177/191) and all instances of it were serotype Ia. The remaining 7 serotype Ia strains were ST103 (clonal complex, CC103) (*n* = 2), ST314 (CC314) (*n* = 2), and ST931 (CC391) (*n* = 1) (Figure 2, Table 4). Two serotype V strains of fGBS were ST931 (strain: CUHK_GBSf18_918) and the locus variant of ST1 (strain: CUHK_fGBS802A_18), which was isolated from bighead carp and tilapia, respectively. The serotype III fGBS strain (strain: 212-CUHK_fGBS22A_17) was a triple-locus variant (TLV) of ST862. The four serotype NT strains were ST7. Serotype Ia ST931 (strain: 224-CUHK_fGBS34A_17) and serotype III-NT TLV of ST862 (strain: 212-CUHK_fGBS22A_17) were isolated from tilapia (Table 4).

The antimicrobial resistance genes (AMRGs) (Table 5) and virulence factors (Table 6) were assessed in fGBS. As expected, the ST7 strains were clonal and rarely carried AMRGs. The virulence factor profiles were similar among the ST7s. The virulence factors of pilus-island 1 was observed in 49 ST7 strains, while none carried pilus-island 2 genes. The FGBS of the minor ST groups contained an average of two AMRGs that conferred resistance to tetracyclines (*tet*M, *tet*S) and aminoglycosides (*ant*(6)*-Ia*, *sat*4). The serotype III strains contained the most AMRGs. These genes conferred resistance to aminoglycosides (*sat*4, *ant*9), macrolides (*erm*B), and lincosamides (*lsa*E, *lnu*B). Only one strain (CUHK_fGBS802A_18, serotype V, DLV of ST1) contained genes encoding the laminin-binding protein, and both pilus-island 1 and 2.

The molecular characterization of pGBS was also analysed through mPCR and WGS on the serotypes and AMRGs, respectively. The serotyping of pGBS (Table 3) showed a predominance of serotype III non-typeable subtype (III-NT) (56/61, 91.8%), followed by three NT strains and two III-2 strains (Table 3). WGS was performed on 59 representative pGBS isolates, encompassing all serotypes. Of the 59 strains, the prevalent clone was ST651 (*n* = 45), followed by the locus variants of ST651 (CC103, *n* = 7), ST862 (CC485, *n* = 6), and a single strain of ST1 (Table 4, Figure 3). In contrast to fish ST7, where AMRGs were scarce and were only observed more often in the minor ST groups of the cohort, the pGBS genomes contained an average of seven AMRGs (Table 5). AMRGs that conferred resistance to aminoglycoside (*ant*(6)-*Ia, aph*(3)-*III*, *sat*4, *spw*), macrolides (*erm*B, *lnu*B), oxazolidinones (*optr*A), phenicol (*cat*A8), and tetracyclines (*tet*L, *tet*M, *tet*S) were observed. The virulence factors *cfa/cfb* were present in all but four strains, and *hyl*B was present in all but one strain. All but one isolate (CUHK_pGBS86A_18, serotype III, ST1) lacked laminin-binding protein and pilus-island 1 and 2 in their virulence factor profile (Table 6).

## 3. Discussion

This is the first comprehensive study to investigate the distribution and characteristics of GBS in food animals from our local wet markets. There are limited data on GBS surveillance in aquaculture and food animals, although similar data have been extensively reported in healthcare settings. We, therefore conducted this study to investigate the distribution and characteristics of GBS in our daily food chain. This territory-wide surveillance study not only provides an update on the burden of GBS in food products from wet markets but also highlights the possible exposure of the bacteria in the community and the seasonality of GBS in food samples.

Distinct GBS molecular characteristics were observed among the two hosts. The isolation rate of GBS in freshwater fishes and pig offal were 19.3% and 16.89%, respectively. Serotype Ia, ST7 (CC7), was the predominant group in fishes, while serotype III ST651 (CC103) was prevalent in pigs. As far as we could ascertain, the sources of the pigs and fish were either local or imported from Guangzhou, China. Serotype Ia ST7 is common in farmed tilapia [16]. It can potentially lead to meningitis and septicaemia in its host, with high mortality rates, as demonstrated by the mass fish death in Kuwait Bay [17,18,19]. Serotype Ia was also more pathogenic than serotype III in fishes [17]. ST7 can also cause invasive diseases in adults and neonates, although it can colonize the vaginal tract as a commensal [18,19]. Our ST7 strains rarely carried AMRG in their genomes and corresponded well to their MIC phenotypes, showing susceptibility to our tested antibiotics. Other STs found in our fish GBS strains included ST103, ST931, ST314, ST1, and TLV862.

In contrast, to fish GBS, GBS in pigs has not been emphasized previously [20]. Our investigation is the first study in two decades to explicitly include pigs for GBS sampling [20]. Pigs, like bovine, can suffer from mastitis caused by GBS. On the other hand, pork is an essential part of Hong Kong’s food culture. Thus, the zoonotic potential is possible through GBS contamination during food processing and handling if proper hygiene is not maintained, although pork is generally fully cooked. ST651 was reported in our previous study, which investigated GBS not susceptible to penicillin and was also observed in pregnant women in mainland China [14,21]. Interestingly, in Southern China, ST862 (9.4% of 266 GBS isolates) was the next most prevalent ST in GBS-positive pregnant women after ST19 [22]. Multidrug resistance (MDR) was observed in 56/61 pGBS isolates from antibiotic susceptibility tests, and multiple AMRGs were found in the pGBS genomes, which confirmed this phenotype. According to the current study, fGBS serotype Ia was less antibiotic-resistant; this correlation was also detected by Chu et al. study [23]. However, the emergence of fluoroquinolone was also detected by Chu et al., whereas in our study, all the isolates were sensitive to fluoroquinolones.

The study of biofilms in medicine currently requires a translational approach, with an examination of clinically relevant biofilms in the context of specific anatomic sites, host tissues, diseases, and genomic evidence used for the quick and confident detection of biofilm-associated infections. However, genes associated with pilus island 1 and 2 biofilm formation were almost absent in the pGBS of the current study.

The MDR GBS observed in pig offal signals the potential of antibiotic resistance in humans and maybe a source of AMR. Furthermore, the AMR levels in pGBS were higher compared to the fGBS in our study. This discrepancy might have been due to the recently introduced food-related initiatives under the Hong Kong strategy and Action Plan on antimicrobial resistance (AMR) and licensing control of livestock keeping, regulating the feeding of drugs and chemicals to food animals, from 2017 [24]. Under this regulation, the application to animals of seven chemicals, namely avoparcin, clenbuterol, chloramphenicol, dienestrol, diethylstilbestrol, hexestrol, and salbutamol, including two antibiotics, was prohibited. However, the use of chemicals, including 36 antibiotics, was restricted for animals in order to address concerns over proper antibiotics usage and the exceeding of drug residue levels for food safety purposes and AMR issues in Hong Kong [25]. Freshwater fishes, such as carp, are staple food sources in China, and they are generally fully cooked during preparation. However, undercooked fish may pose a risk of GBS contamination. We previously reported its association with a higher prevalence of human GBS ST283 disease during summer [26]. This also coincides with the higher rates of fish GBS strains when the temperature is the highest and with heavy rainfall. This phenomenon was also observed in our experiment on fGBS from freshwater fish. Seasonality in pGBS isolation was inconclusive since only a year of speculation was performed. Although we reported ST283 disease in Hong Kong, it was only sporadically isolated per year from one of the cluster hospitals. Recent studies demonstrated that ST283 was prevalent in fish farms in Southeast Asia and was a predominant ST type in GBS bacteraemia from Thailand, Laos, and Vietnam [8]. No ST283 was detected in our food animals, which raises the possibility that our patients may have acquired the clone from elsewhere, such as when travelling to countries in Southeast Asia and consuming local delicacies with a high risk of GBS ST283 contamination. Through communications with the vendors, we ensured that the freshwater fishes from our study were less likely to have been imported from Southeast Asian countries, due to high locomotive and shipment costs. Hence, the travel history of patients, consumption, or contact with freshwater fish should be ascertained from hospitalized patients with GBS sepsis in the future.

## 4. Materials and Methods

### 4.1. Animal Sampling

Nine-hundred-and-ninety-two fish were purchased from the wet markets of Hong Kong, covering three geographical regions (Kowloon, the New Territories, and Hong Kong Island) across 18 districts between June 2016 and September 2019. Among the 992 fish samples, 805 were whole tilapia fish (*Oreochromis mossambicus*) (*n* = 182), snakehead fish (*Ophicephalus maculatus*) (*n* = 314), and black carp (*Mylopharyngodon piceus*) (*n* = 309), while 187 were heads of bighead carp (*Hypophthalmichthys nobilis*). In addition, three-hundred-and-sixty-one pig offal were procured from the wet markets between April 2018 and March 2019. Samples were transported in cooler bags and processed on the day.

### 4.2. Isolation of GBS

The skin, flesh, gills, and internal organs (namely the hearts, spleens, livers, and guts) of freshwater fish were obtained in a sterile manner to avoid handling and environmental contamination, whereas only skin, gills, and flesh of bighead carps were taken. Deep tissues of pig offal (namely tongues and large and small intestines) were dissected. It could not be confirmed whether one set of offal (i.e., tongue, large and small intestine) was derived from a single pig. A small piece of tissue weighing a maximum of 10 g from representative body sites was individually transferred to 50 mL Todd-Hewitt broth (Thermo Fisher Scientific, Hampshire, UK) supplemented with four micrograms per millilitre of polymyxin B and 16 µg/mL nalidixic acid in a stomacher (Daigger Scientific, Hamilton, NJ, USA) for 30 min prior to overnight incubation at 37 °C for primary GBS enrichment [27]. Secondary enrichment for GBS growth was performed by incubating 1% culture in 5 mL of brain–heart infusion broth (BHI) (Thermo Fisher Scientific, Hampshire, UK) overnight. The enriched broth was plated on ChromID STREPTB agar (BioMerieux, Marcy l’Etoile, France) for overnight culture. A maximum of ten magenta colonies per body site was picked and confirmed by either MALDI-TOF, Streptex test (Thermo Fisher Scientific, Branchburg, NJ, USA), or GBS-specific PCR before further characterization [28]. If multiple organs of a fish were confirmed with GBS colonization, the selection of representative strains for downstream analysis was prioritized in the internal organs (i.e., the heart, spleen, liver, flesh, and gut), and then in the external organs (i.e., the gills and skin). Details of the food animals, including their origin, purchase location, types of samples, and status (e.g., bruises, ulcers, or other damages) were recorded. All confirmed GBS strains were stored at −80 °C in 10% (*v*/*v*) glycerol-BHI broth for further experiments.

### 4.3. Antibiotic Susceptibility Testing

Antibiotic susceptibility test on 11 antibiotics was carried out on GBS strains via micro broth dilution, according to CLSI, with *Streptococcus pneumoniae* ATCC49619 as control [15]. The antibiotics used in this study were ciprofloxacin (CIP), levofloxacin (LEV), gentamicin (GEN), tetracycline (TET), minocycline (MIN), doxycycline (DOX), clindamycin (CLI), linezolid (LNZ), erythromycin (ERY), penicillin (PEN), and vancomycin (VAN). MIC_50_ and MIC_90_ were detected for the comparison of resistance.

### 4.4. Biofilm Formation

Biofilm formation in fish and pig GBS was performed using the tissue culture plate method, as described previously, with slight modifications [29]. GBS isolates were incubated aerobically for 24 h at 37 °C with tryptic soy broth (TSB) (Thermo Fisher Scientific, Hampshire, UK) containing 1% glucose prior to 1:100 dilution with TSB. Triplicate wells with broth only served as a control to check the non-specific binding of media. Two hundred microliters of diluted culture were inoculated in triplicates in sterile 96-well flat-bottomed tissue culture plates for 24 h at 37 °C. The supernatant was removed by gently tapping on the plates. Wells were washed three times with phosphate buffer saline (pH 7.2) to remove free-floating bacteria. Biofilms formed by adherent sessile organisms were fixed with 4% formaldehyde for 20 min. Formaldehyde was removed, and the plate was left for drying. All wells were stained with crystal violet (0.1% *v*/*v*), and any excess stain was thoroughly rinsed off with deionized water and left to dry. A freshly prepared 95% ethanol mixture (200 µL) was added to dissolve bounded crystal violet. Density results were interpreted as high, moderate, or non-biofilm-forming at mean OD_595nm_ thresholds of 0.24 and 0.12, according to Abdul-Lateef et al., 2018 [29]. One-way ANOVA was performed on biofilm formation for comparison among the two hosts.

### 4.5. Molecular Characterization of GBS Isolates and Whole-Genome Sequencing

Molecular characterization of GBS strains was performed through serotyping and whole-genome sequencing. Whole-genome sequencing was performed on one-hundred-and-ninety-one fish GBS genomes; four serotype Ia strains failed quality control and were removed from the genome analysis. In the same way, WGS was performed on sixty-one pig GBS isolates; two were removed from the genome analysis due to poor quality after repeating the experiment. According to the manufacturer’s protocol, GBS strains underwent DNA extraction for whole-genome sequencing (WGS) (Wizard^®^ Genomic DNA purification kit (Promega, Madison, WI, USA). Serotyping for capsular polysaccharide antigens I-IX was performed as previously described [30]. Isolates that failed to be assigned a serotype were grouped as non-typeable (NT).

Library preparation was performed using a Riptide High-Throughput Rapid DNA library preparation kit (iGenomix, Jersey City, NJ, USA) according to the manufacturer’s instruction. Genomes were sequenced by NextSeq mid output 500, obtaining pair-end reads at 150 bp (Illumina, San Diego, CA, USA). Sequence reads were demultiplexed according to the manufacturer’s instructions as to the library preparation kit prior to our genome assembly pipeline, as previously described [30]. Briefly, the de novo assembly of the sequence reads was generated by SPAdes (3.10.1) [31], where contigs with depths < 5 and lengths < 500 bp were filtered. Resistant gene profiles were acquired by blasting and read-mapping to ResFinder [32]. Virulence factors were identified using VFDB [33]. PubMLST database was used for multi-locus sequence typing (MLST) [34]. A pangenome tree was constructed with PARSNP and visualized with iTOL [35,36]. Sequence raw reads of the isolates are available from NCBI BioProject (No.: PRJNA52017).

## 5. Limitations

The availability of Tilapia is seasonal in Hong Kong, where supply diminishes in autumn and winter; thus, other freshwater fishes commonly used in Hong Kong cuisine, snakehead and black carp, were also included in the study. Serotype III was not identified in our fish strains; thus, we could not associate serotype III ST283 and its infection in Hong Kong, unlike in Singapore. Instead, other serotypes, Ia in fish and serotype III-NT in pigs, were found, while MDR was observed in the pGBS. Since pork is commonly used in Chinese cuisine and serotype III with MDR was frequently found in our study on pig offal, one may postulate that the handling of pork and poor hygiene may substantiate the potential of zoonotic serotype III GBS infection in the community.

## 6. Conclusions

This is the first comprehensive report on the prevalence and molecular characteristics of *Streptococcus agalactiae* in a locality. Serotype Ia, ST7, and serotype III ST651 were the predominant GBS strains in our food animals from wet markets. Antimicrobial resistance was scarce in the fish GBS, but multidrug resistance was observed in the porcine GBS. As pork is a staple food in Hong Kong and China, antimicrobial resistance can spread from a One Health perspective. Thus, policies on the use of antibiotics in pig farming should be reviewed.

## Figures and Tables

**Figure 1 antibiotics-11-00397-f001:**
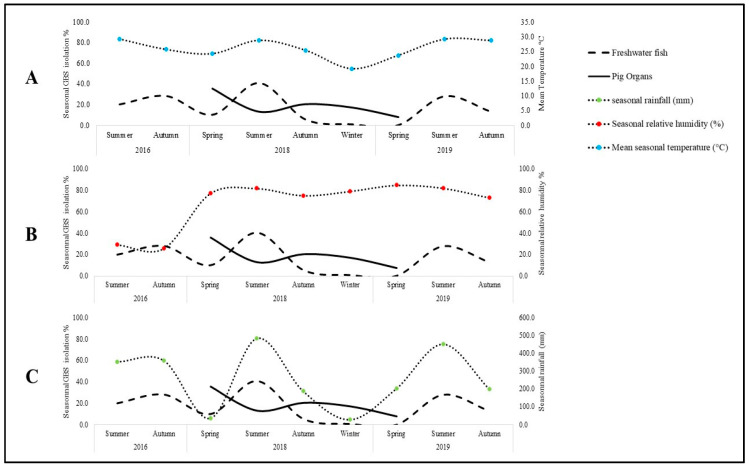
Association of climate and prevalence of GBS in fish and pig offal in Hong Kong. Percentage of GBS isolation among fish samples and pig organs in relation to (**A**) mean seasonal temperature, (**B**) seasonal relative humidity, and (**C**) seasonal rainfall is shown.

**Figure 2 antibiotics-11-00397-f002:**
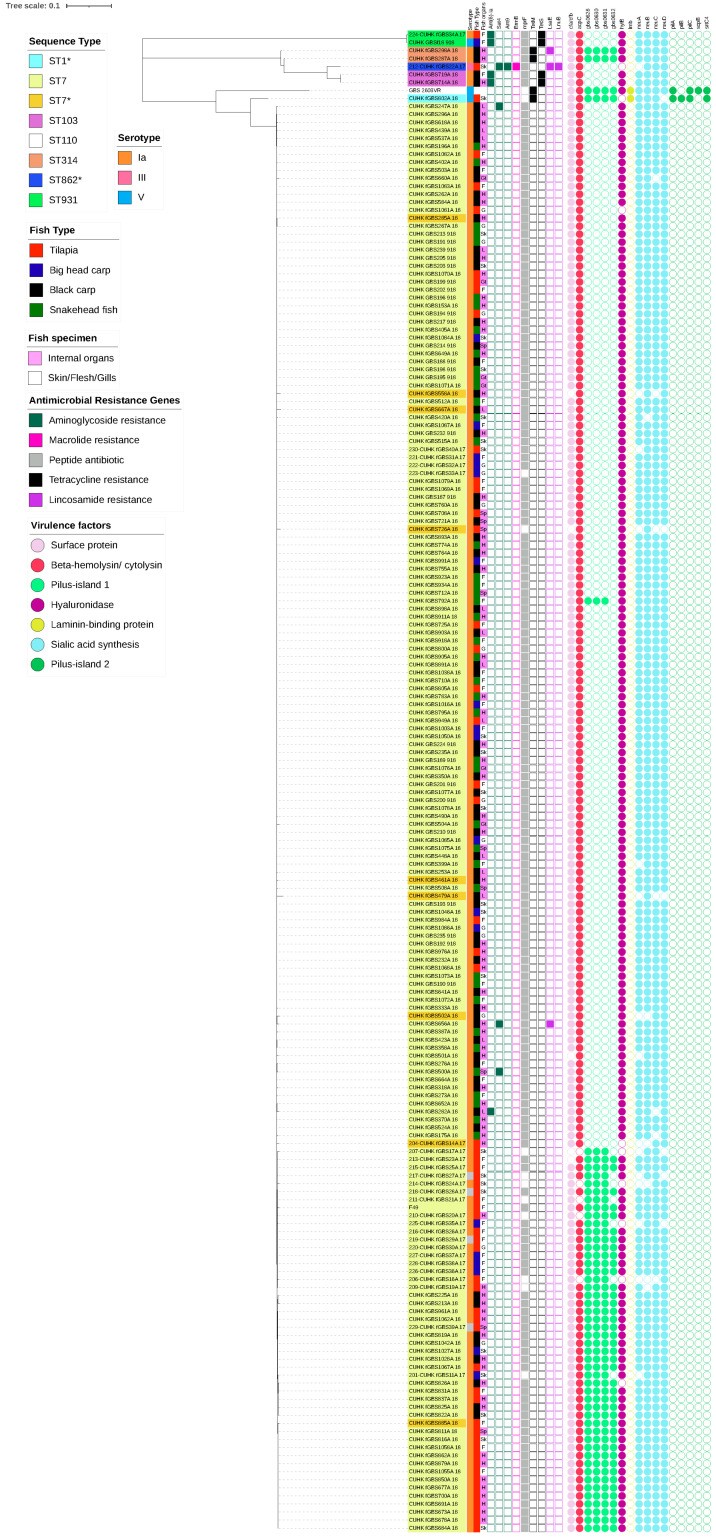
Phylogeny of fish GBS and profiles’ molecular characteristics. Coloured ribbons on the labels indicate the sequence type (ST), while the coloured strip adjacent to the labels shows the serotypes. The type of fish and specimen sampled are also noted adjacent to the serotype. Antimicrobial resistance genes and virulence factors are noted in squared boxes and circles, respectively. Sequence types with an asterisk (*) indicate strains with locus variants of the sequence types (SLV/DLV/TLV).

**Figure 3 antibiotics-11-00397-f003:**
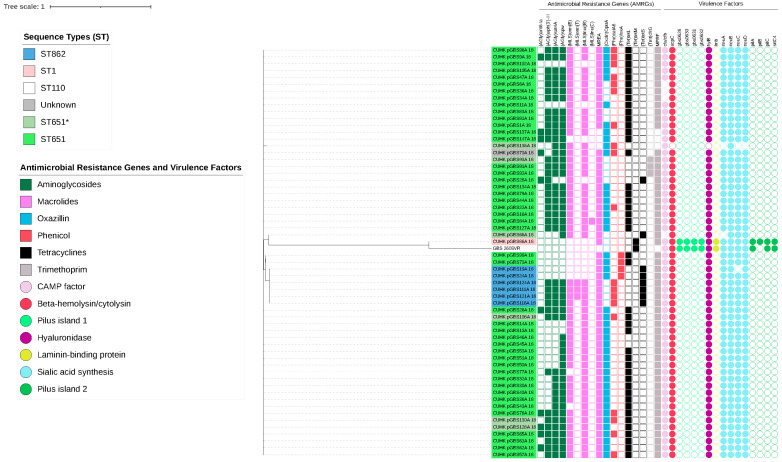
Phylogeny of pig GBS profiles’ molecular characteristics. Coloured ribbons on the labels indicate the sequence type; antimicrobial resistance genes and virulence factors are noted in squared boxes and circles, respectively. Sequence types with an asterisk (*) indicate strains with locus variants of the sequence types (SLV/DLV/TLV).

**Table 1 antibiotics-11-00397-t001:** Prevalence of GBS in freshwater fish and pig samples procured from wet markets in Hong Kong.

	Fish Type/Food Source (*n*) *	GBS Prevalence %, (*n*)
Freshwater fish (*n* = 191/992, 19.3%)	Tilapia (*n* = 182)	34.1% (62)
	Big Head Carp (*n* = 187)	10.1% (19)
	Snakehead Fish (*n* = 314)	13.6% (43)
	Black Carp (*n* = 309)	22.3% (67)
Pigs (*n* = 61/361, 16.9%)	Tongue (*n* = 193)	24.8% (48)
	Small intestine (*n* = 92)	3.2% (3)
	Large intestine (*n* = 76)	13% (10)

* Number of fish samples or pig offal collected in the experiment.

**Table 2 antibiotics-11-00397-t002:** Minimum inhibition concentrations (MICs) of GBS against 11 antibiotics.

Class of Antibiotic	Antibiotic	Concentration Range (μg/mL)	Freshwater Fish GBS			Pig GBS		
			MIC (mg/L) ^a^		% (No./191) Resistance	MIC (mg/L) ^a^		% (No./61) Resistance
			MIC_50_	MIC_90_		MIC_50_	MIC_90_	
Penicillins	Penicillin	2–0.0625	0.03	0.06	0.5 (1)	0.015	0.03	0 (0)
Glycopeptides	Vancomycin	8–0.25	0.25	0.5	0 (0)	0.12	0.25	0 (0)
Tetracyclines	Doxycycline	32–0.12	0.25	2	37.1 (71)	16	16	90.1 (55)
	Minocycline	32–0.12	≤0.12	1	5.7 (11)	16	16	85.2 (52)
	Tetracycline	32–0.12	≤0.12	1	5.7 (11)	16	16	90.1 (55)
Oxazolidinones	Linezolid	64–0.06	2	2	0 (0)	2	2	0 (0)
Macrolides	Erythromycin	4–0.12	≤0.12	≤0.12	3.1 (6)	>16	>16	88.5 (54)
Lincosamides	Clindamycin	4–0.12	≤0.12	≤0.12	1.5 (3)	>16	>16	98.3 (60)
Fluoroquinolones	Ciprofloxacin ^^^	32–0.12	0.5	1	0.5 (1)	0.5	0.5	0 (0)
	Levofloxacin	32–0.12	0.25	0.5	0 (0)	0.5	0.5	0 (0)

MIC breakpoints for GBS were referenced according to 2019 CLSI guidelines [15]. ^a^ Breakpoints for defining sensitive strains (in mg/L) of the following antibiotics are in parentheses: CIP ciprofloxacin (≤1 mg/L); LEV, levofloxacin (≤2 mg/L); GEN, gentamicin (≤1 mg/L); TET, tetracycline (≤2 mg/L); MIN, minocycline (≤2 mg/L); DOX, doxycycline (≤2 mg/L); PEN, penicillin (≤0.12 mg/L); CLI, clindamycin (≤0.25 mg/L); LNZ, Linezolid (≤2 mg/L); VAN, vancomycin (≤1 mg/L); ERY, erythromycin (≤0.25 mg/L). ^^^ CLSI breakpoint for *Enterococcus* spp. was used.

**Table 3 antibiotics-11-00397-t003:** Serotype distribution of *S. agalactiae* isolated from freshwater fish and pig organs.

Food Type	No. ^a^	Serotypes *n* (%)				
		Ia	III-2	III-NT	V	NT
Tilapia	62 ^#^	56	-	1	1	4
		(90.3)		(1.6)	(1.6)	(6.3)
Big Head cap	19	18	-	-	1	-
		(94.7)			(5.3)	
Snakehead	43	43	-	-	-	-
		(100)				
Black carp	67	67	-	-	-	-
		(100)				
Pig’s tongue	48	-	2	45	-	1
			(4.2)	(93.8)		(2.1)
Pig’s small intestine	3	-	-	3	-	-
				(100)		
Pig’s large intestine	10	-	-	8	-	2
				(80)		(20)
Total GBS strains collected	252	184	2	57	2	7
		(73)	(0.8)	(22.6)	(0.8)	(2.8)

^a^ Number of GBS collected from each food type. NT—non-typeable according to PCR protocol. III-NT—serotype III non-subtypeable. ^#^ Two GBS strains belonging to serotype Ia and III-NT were recovered from tilapia number T25.

**Table 4 antibiotics-11-00397-t004:** Distribution of STs among fish and pig GBS.

Source	Number of GBS Strains	Sequence Types (STs) *n* (%)									
		1	7	SLV7	103	314	651	SLV651	862	931	Unknown ^#^
Fish	191	1 (0.4)	175 (69.4)	9 (3.6)	2 (0.8)	2 (0.8)	-	-	-	2 (0.8)	-
Pig	61	1 (0.4)	-	-	-	-	47 (18.7)	6 (2.4)	6 (2.4)	-	1 (0.4)
Total	252	2 (0.8)	175 (69.4)	9 (3.6)	2 (0.8)	2 (0.8)	47 (18.7)	6 (2.4)	6 (2.4)	2 (0.8)	1 (0.4)

*n* = number of GBS strains belonging to each ST. ^#^ ST was not identifiable according to the genome data.

**Table 5 antibiotics-11-00397-t005:** Antimicrobial-resistant genes available among fish and pig GBS.

Antibiotic Group	Resistant Gene	Fish		Pig	
		No. of Strains Carrying the Gene (*n* = 191)	Percentage (%)	No. of Strains Carrying the Gene (*n* = 61)	Percentage (%)
Aminoglycoside	*ant*(6)-*Ia*	5	2.6	10	16.4
	(*AGly*)*aph*(3′)-*II*	-	-	37	60.7
	*sat*4	3	1.6	42	68.9
	(*Agly*)*spw*	-	-	48	78.7
Peptide antibiotic	*mpr*F	179	93.7	-	-
Tetracyclines	*tet*L	-	-	43	70.5
	*tet*M	4	2.1	2	3.3
	*tet*S	4	2.1	8	13.1
Lincosamide	*lsa*E	2	1.0	-	-
Macrolides	*erm*B	-	-	50	82.0
	*erm*T	-	-	3	4.9
	*erm*B *+ erm*T	-	-	43	70.5
	*lnu*B	-	-	49	80.3
	*lnu*C	-	-	1	1.6
	*inu*B *+ lnu*C	-	-	1	1.6
	*MREA*	-	-	55	90.2
Oxacillin	*optr*A	-	-	43	70.5
Phenicol	*cat*A8	-	-	19	31.1
	*fex*A	-	-	4	6.6
Trimethoprim	*dfr*G	-	-	3	4.9
	*MPRF*	-	-	55	90.2

**Table 6 antibiotics-11-00397-t006:** Distribution of virulence genes between fish and pig GBS strains.

Category	Virulence Gene	Total No. of Isolates with Virulence Genes (%)	
		Fish GBS (*n* = 191)	Pig GBS (*n* = 61)
Adhesion	*Lmb*	2 (1.0)	2 (3.3)
	*gbs*0628 ^a^	53 (27.7)	2 (3.3)
	*gbs*0630 ^a^	53 (27.7)	2 (3.3)
	*gbs*0631 ^a^	53 (27.7)	2 (3.3)
	*gbs*0632 ^a^	45 (23.6)	2 (3.3)
	*pil*A	2 (1.0)	2 (3.3)
	*pil*B	1 (0.5)	1 (1.6)
	*pil*C	2 (1.0)	2 (3.3)
	*scp*B	1 (0.5)	0 (0.0)
	*sr*C4	2 (1.0)	2 (3.3)
Invasion	*cyl*A	183 (95.8)	59 (96.7)
	*cyl*B	184 (96.3)	59 (96.7)
	*cyl*D	179 (93.7)	59 (96.7)
	*cyl*F	185 (96.9)	58 (95.1)
	*cyl*G	188 (98.4)	59 (96.7)
	*cyl*I	184 (96.3)	57 (93.4)
	*cyl*J	182 (95.3)	58 (95.1)
	*cyl*K	181 (94.8)	58 (95.1)
	*cyl*L	0 (0.0)	0 (0.0)
	*cyl*R1	0 (0.0)	0 (0.0)
	*cyl*R2	0 (0.0)	0 (0.0)
	*cyl*S	0 (0.0)	0 (0.0)
	*cyl*X	187 (97.9)	59 (96.7)
	*cyl*Z	190 (99.5)	59 (96.7)
	*acp*C	186 (97.4)	58 (95.1)
	*hyl*B	183 (95.8)	58 (95.1)
Immune evasion	*cps*A	173 (90.6)	58 (95.1)
	*cps*B	183 (95.8)	59 (96.7)
	*cps*C	180 (94.2)	58 (95.1)
	*cps*D	0 (0.0)	59 (96.7)
	*cps*E	180 (94.2)	56 (91.8)
	*cps*F	183 (95.8)	59 (96.7)
	*cps*G	2 (1.0)	2 (3.3)
	*cps*H	3 (1.6)	2 (3.3)
	*cps*I	0 (0.0)	0 (0.0)
	*cps*J	3 (1.6)	2 (3.3)
	*cps*K	166 (86.9)	58 (95.1)
	*cps*L	178 (93.2)	58 (95.1)
	*cps*M	3 (1.6)	2 (3.3)
	*cps*N	3 (1.6)	0 (0.0)
	*cps*O	3 (1.6)	2 (3.3)
	*neu*A	177 (92.7)	58 (95.1)
	*neu*B	184 (96.3)	59 (96.7)
	*neu*C	184 (96.3)	57 (93.4)
	*neu*D	186 (97.4)	58 (95.1)

^a^ GBS pilus cluster.

## Data Availability

All data generated or used during the study appear in the submitted article or Appendix A.

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
