# Peer review of "Prevalence and Characteristics of Streptococcus agalactiae from Freshwater Fish and Pork in Hong Kong Wet Markets"

_antibiotics, 2022, doi:10.3390/antibiotics11030397_

Round 1

Reviewer 1 Report

This manuscript explored the characteristics and distribution of GBS in fish and pork. They showed general characteristics of GBS including serotypes, sequence types (ST), antimicrobial resistance, antimicrobial resistance genes (AMR). Overall of this manuscript is interesting, however, additional detail should be improved. I have some comments appeared below.

  1. Abstract (line 12-13). It should mention how many GBS detected in each fish and pig.?
  2. Introduction (line 39). GBS is not caused disease in newborn or pregnant, it also caused invasive infection in adult. So, the author can add the sentence similar like this "GBS can cause an invasive infection in adults", please see (Paveenkittiporn et al. Streptococcus agalactiae infections and clinical relevance in adults, Thailand. Diagn Microbiol Infect Dis. 2020 May;97(1):115005. doi: 10.1016/j.diagmicrobio.2020.115005.) as the reference for cite.
  3. Introduction (lines 70-76). What is idea/background/pain point to select fish and pork as the samples to study?
  4. Introduction, line 71 "....the Singapore outbreak", reference is needed to cite.
  5. Table 2, Line 121-123: (1) CLSI breakpoint for Enterococcus is not recommend or use to interpret susceptibility of GBS (it is beta-hemolytic Streptococcus/pyogenic streptococcus). The author should  apply the breakpoint for beta hemolytic group Streptococcus of CLSI. So, interpretation of antimicrobial susceptibility in this study should be revised. This is my major concern. (2) CLSI version 2017 is too old, at least 2019 is required because this study period is 2016-2019.
  6. I recommend the author to create additional Table to summarize WGS analysis including serotype, STs, AMR genes, virulence genes for fGBS and pGBS in the result part.
  7. Figure 2 & 3 can be combined because all of them were GBS. It should include in the same tree.
  8. Result part: The authors should show or describe about AMR genes and susceptibility. And this result must be mentioned in discussion part.
  9. Discussion (line 203-207): Deep detail of AMR discussion need to be mentioned. The author must compare your susceptibility result to another literature.
  10. Discussion: pGBS revealed more resistance to antibiotics and AMR genes than fGBS, this point should be discussed that why?
  11. Discussion. In this study the author found serotype III is a major serotype prevalent in pork. Another study of human infection also found serotype III is mainly serotype for invasive infection (Paveenkittiporn et al. Streptococcus agalactiae infections and clinical relevance in adults, Thailand. Diagn Microbiol Infect Dis. 2020 May;97(1):115005. doi: 10.1016/j.diagmicrobio.2020.115005), so, the author can mention and refer to the important of serotype III virulence or pathogenesis.
  12. Discussion: No discussion about biofilm, what is importance? Is biofilm related to what?
  13. M&M: I do not understand for sample preparation from fish. Line 241, you mentioned about 10 g tissue was macerated in 50 ml of THB. So, how are you prepare from 1 fish? because you taked skin, flesh and internal organs (line 236). All of them were taken only 10 g or take 10 g for each specimen sites from 1 fish.  I think that an additional  section of "sample preparation" should be added. The author need to explain deep detail about sample preparation, especially from fish.
  14. M&M. How many colonies you picked up per sample?
  15. M&M (lines 256-262). CLSI should be update, at least version 2019 and interpretation breakpoint of beta-hemolytic Streptococcus should be used.
  16. M&M (Lines 280-285): How many isolates from pig and fish applied for WGS?
  17. M&M (lines 295-298): What program do you used to determine AMR genes?

Author Response

Responses to comments from the reviewers

The authors are very grateful for the reviewers for their valuable time and constructive comments which help improve the quality and clarity of our manuscript.  We would like to address their comments point-by-point below.

Reviewer 1:

This manuscript explored the characteristics and distribution of GBS in fish and pork. They showed general characteristics of GBS including serotypes, sequence types (ST), antimicrobial resistance, antimicrobial resistance genes (AMR). Overall, of this manuscript is interesting, however, additional detail should be improved. I have some comments appeared below.

  1. Abstract (line 12-13). It should mention how many GBS detected in each fish and pig.?

We detected 191 fish GBS and 61 pig GBS during this study period and it has now been incorporated to line 11.

  1. Introduction (line 39). GBS is not caused disease in newborn or pregnant, it also caused invasive infection in adult. So, the author can add the sentence similar like this "GBS can cause an invasive infection in adults", please see (Paveenkittiporn et al. Streptococcus agalactiae infections and clinical relevance in adults, Thailand. Diagn Microbiol Infect Dis. 2020 May;97(1):115005. doi: 10.1016/j.diagmicrobio.2020.115005.) as the reference for cite.

The reference has been incorporated to line 39

  1. Introduction (lines 70-76). What is idea/background/pain point to select fish and pork as the samples to study?
  2. Both West Pacific and East Asian countries are significant producers and consumers of freshwater fish, and consumption of raw fish is common in the region, we hypothesize that there is potential zoonosis acquired from fish, as in the Singapore outbreak. This was mentioned in line 70-73.
  3. Introduction, line 71 "....the Singapore outbreak", reference is needed to cite.

Reference cited in the text

Tan S, Lin Y, Foo K, Koh HF, Tow C, Zhang Y, Ang LW, Cui L, Badaruddin H, Ooi PL, Lin RTP, Cutter J. Group B streptococcus serotype III sequence type 283 bacteremia associated with consumption of raw fish, singapore. Emerg Infect Dis . 2016 11 [cited Dec 15, 2020];22(11):1970-3

  1. Table 2, Line 121-123: (1) CLSI breakpoint for Enterococcus is not recommend or use to interpret susceptibility of GBS (it is beta-hemolytic Streptococcus/pyogenic streptococcus). The author should  apply the breakpoint for beta hemolytic group Streptococcus of CLSI. So, interpretation of antimicrobial susceptibility in this study should be revised. This is my major concern. (2) CLSI version 2017 is too old, at least 2019 is required because this study period is 2016-2019.

The CLSI version has been updated to 2019 and the relevant Tables have been revised accordingly. We have taken CLSI breakpoints of beta-hemolytic Streptococcus for most of the antibiotics, except for doxycycline, ciprofloxacin and minocycline where it was not available; thus, Enterococcus was used for these three antibiotics. This has been noted under the MIC table.  

  1. I recommend the author to create additional Table to summarize WGS analysis including serotype, STs, AMR genes, virulence genes for fGBS and pGBS in the result part.

Table numbers 3, 4, 5 and 6 were incorporated to the Results section under serotype distribution, ST distribution, AMR genes and virulence factor distribution among fish and pig GBS respectively.

  1. Figure 2 & 3 can be combined because all of them were GBS. It should include in the same tree.

Owing to the concern of the visual clarity upon publication if the two figures were combined to one, we think that retaining the two figures as separate would be more appropriate. Figure 2 illustrates strains of Serotype Ia ST7 fish GBS while Figure 3 describes those of pork GBS. Furthermore, having two separate phylogeny figures would be more consistent to our flow of tables and results where we kept the numbers and findings of the two GBS hosts separate yet comparable.

  1. Result part: The authors should show or describe about AMR genes and susceptibility. And this result must be mentioned in discussion part.

AMR and susceptibilities are mentioned in line 168-170 under Results section and in lines 227-230 in Discussion section.

  1. Discussion (line 203-207): Deep detail of AMR discussion need to be mentioned. The author must compare your susceptibility result to another literature.

Significant detection of penicillin-non-susceptible fish GBS strain in our study was explained and compared in lines 227-230 and compared to the Penicillin non-susceptible strain details as previously published.

Reference: Li C, Sapugahawatte DN, Yang Y, Wong KT, Lo NWS, Ip M. Multidrug-resistant streptococcus agalactiae strains found in human and fish with high penicillin and cefotaxime non-susceptibilities. Microorganisms. 2020 July 16;8(7):10.3390/microorganisms8071055. doi: 10.3390/microorganisms8071055.

  1. Discussion: pGBS revealed more resistance to antibiotics and AMR genes than fGBS, this point should be discussed that why?

The possible reason to this AMR issue has been addressed in lines 236-247

  1. Discussion. In this study the author found serotype III is a major serotype prevalent in pork. Another study of human infection also found serotype III is mainly serotype for invasive infection (Paveenkittiporn et al. Streptococcus agalactiae infections and clinical relevance in adults, Thailand. Diagn Microbiol Infect Dis. 2020 May;97(1):115005. doi: 10.1016/j.diagmicrobio.2020.115005), so, the author can mention and refer to the important of serotype III virulence or pathogenesis.

The prevalence of Serotype III ST651 from pig GBS have been noted in current study and it was mentioned in the Discussion in line 206. The comparison of serotype III with available literature has also been incorporated in the Discussion in lines 206-211with the pathogenesis and virulence of GBS serotype III mentioned.

  1. Discussion: No discussion about biofilm, what is importance? Is biofilm related to what?

Discussion about biofilm formation has been incorporated in lines 231-235 (under Discussion).

  1. M&M: I do not understand for sample preparation from fish. Line 241, you mentioned about 10 g tissue was macerated in 50 ml of THB. So, how are you prepare from 1 fish? because you taked skin, flesh and internal organs (line 236). All of them were taken only 10 g or take 10 g for each specimen sites from 1 fish.  I think that an additional section of "sample preparation" should be added. The author need to explain deep detail about sample preparation, especially from fish.

The details have been rephrased for clarity in Line numbers 281-283.

  1. M&M. How many colonies you picked up per sample?

A maximum of ten colonies were picked per sample site. This detail was incorporated in lines 288-289.

  1. M&M (lines 256-262). CLSI should be update, at least version 2019 and interpretation breakpoint of beta-hemolytic Streptococcus should be used.

Please refer to the point number five reply.

  1. M&M (Lines 280-285): How many isolates from pig and fish applied for WGS?

One hundred and eighty-seven fish GBS strains and 59 pig GBS strains were subjected to WGS analyses and the details were explained in line 324-327.

  1. M&M (lines 295-298): What program do you used to determine AMR genes?

Resistant gene profiles were acquired by blasting and read mapping to ResFinder and reference was incorporated in line 340-341

Reviewer 2 Report

The manuscript entitle "Prevalence and characteristics of Streptococcus agalactiae from Freshwater Fish and Pork in Hong Kong Wet Markets" describes the antibiotics resistant the serotype identification and other phenotype characteristics of strains isolates from a local market.

The flow of the paper is very good and the results clearly expressed.

Minor comments:

Line 16: 19,3% plus 16,9% is equal to more than 36% but what happened with the 64%???

Line 36: please microbiota not flora... 

Lines 39-40 and 42-44: It sound like the same idea, I know that the first lines is about the host and the other about economically importance host but It sounds repetitive, Could the author mix both sentence in one??

Lines 64-65: could the authors incorporate this sentence to the before (lines 62-63) please? something like ....beef, pork, included offal, head, ....

Lines 229-231: Please separate the 805 samples, how many for tilapia, snakehead fish and carp separately.

Author Response

Reviewer 2

The manuscript entitle "Prevalence and characteristics of Streptococcus agalactiae from Freshwater Fish and Pork in Hong Kong Wet Markets" describes the antibiotics resistant the serotype identification and other phenotype characteristics of strains isolates from a local market.

The flow of the paper is very good and the results clearly expressed.

Minor comments:

Line 16: 19,3% plus 16,9% is equal to more than 36% but what happened with the 64%???

We have calculated the isolation rate separately for pig and fish. The isolation rate of fish GBS was 19.3% (191 strains from 992 fresh water fish), and pig GBS was 16.9% (61 strains from 361 pig organs). Therefore, 801/992 fish and 300/361 pig organs were GBS negative during our investigation This description has been rephrased in Lines 16-17.

Line 36: please microbiota not flora... 

The word ‘flora’ was replaced by ‘microbiota’ as suggested in line 36.

Lines 39-40 and 42-44: It sound like the same idea, I know that the first lines is about the host and the other about economically importance host but It sounds repetitive, Could the author mix both sentence in one??

This sentence has been rephrased in line numbers 42-46 for clarity.

Lines 64-65: could the authors incorporate this sentence to the before (lines 62-63) please? something like ....beef, pork, included offal, head, ....

The wordings have been clarified in lines 61 to 70.

Lines 229-231: Please separate the 805 samples, how many for tilapia, snakehead fish and carp separately.

Among the 992 fish samples, 805 of them were whole fish of tilapia (Oreochromis mossambicus) (n=182), snakehead fish (Ophicephalus maculatus) (n=314) and black carps (Mylopharyngodon piceus) (n=309), while 187 were heads of bighead carps (Hypophthalmichthys nobilis). This detail has been added in line 271-273.

Round 2

Reviewer 1 Report

None